

# Dynamic guided metric representation learning for multi-view clustering

Tingyi Zheng[1,2], Yilin Zhang[3] and Yuhang Wang[4]

[1] College of Information and Computer, Taiyuan University of Technology, Taiyuan, Shanxi, China
[2] Department of Electrical and control Engineering, Shanxi Institute of Energy, Jinzhong, Shanxi, China
[3] Software College, Taiyuan University of Technology, Taiyuan, Shanxi, China
[4] College of Data Science, Taiyuan University of Technology, Taiyuan, Shanxi, China

## ABSTRACT

Multi-view clustering (MVC) is a mainstream task that aims to divide objects into meaningful groups from different perspectives. The quality of data representation is the key issue in MVC. A comprehensive meaningful data representation should be with the discriminant characteristics in a single view and the correlation of multiple views. Considering this, a novel framework called Dynamic Guided Metric Representation Learning for Multi-View Clustering (DGMRL-MVC) is proposed in this paper, which can cluster multi-view data in a learned latent discriminated embedding space. Specifically, in the framework, the data representation can be enhanced by multi-steps. Firstly, the class separability is enforced with Fisher Discriminant Analysis (FDA) within each single view, while the consistence among different views is enhanced based on Hilbert-Schmidt independence criteria (HSIC). Then, the 1st enhanced representation is obtained. In the second step, a dynamic routing mechanism is introduced, in which the location or direction information is added to fulfil the expression. After that, a generalized canonical correlation analysis (GCCA) model is used to get the final ultimate common discriminated representation. The learned fusion representation can substantially improve multi-view clustering performance. Experiments validated the effectiveness of the proposed method for clustering tasks.

## INTRODUCTION

Multi-view data are extremely common in many applications, each individual view and the correlation of multiple views have their specific property for a particular knowledge discovery task. Multi-view data often contain diversity as well as consistent information that should be exploited and fused. Therefore, considering the uniqueness, complementary, and correlation of each view, it is essential to study how to fuse multi-view data efficiently. Recently, increasing research efforts have been made in multi-view learning, where multi-view clustering (MVC) forms a mainstream task that aims to divide subjects into meaningful groups from different perspectives by learning the multi-view information (*e.g.*, *Yang & Wang, 2018*). However, a better clustering representation can be learned by simultaneously analyzing the discriminant characteristics of a single view and the correlations among

Corresponding author
Tingyi Zheng, tyut66666@163.com

multiple views. Therefore, a priority for most MVC methods is to find a feasible and direct way to explore the underlying data cluster latent fusion representation by multiple views to obtain the final ideal clustering results.

Many research work about MVC have been investigated in the last decades, in which most of them proposed to effectively consider rich information from multiple views (*e.g.*, *Wang, Yang & Liu, 2020*). For instance, co-training (*e.g.*, *Xia, Yang & Yu, 2020*) and co-regularized (*e.g.*, *Kang, Shi & Huang, 2020*) spectral clustering have been proposed to minimize the disagreement between each pair of views (*e.g.*, *Chen, Huang & Wang, 2020*). However, clustering performance is easily affected by the poor quality of original views in this kind of method. Multi-view subspace clustering uses the unified shared feature representation of each view to obtain consistent clustering results from multiple views (*e.g.*, *Huang, Kang & Xu, 2020*; *Zhao, Ding & Fu, 2017a*; *Zhao, Ding & Fu, 2017b*). Typical models also include subspace learning-based and non-negative matrix factorization (NMF)-based models. In the past decade, numerous machine learning technologies have been investigated to determine the scope of combining multiple views. However, these methods lack the ability to mine the latent unified representation and to learn the non-linear correlation of views. To address the second limitation, many researchers proposed multi-kernel and CCA-based clustering methods. The multi-kernel method uses predefined kernels for each view and combines these kernels in linear or non-linear mode. Nevertheless, the complex relationships make it difficult to represent data fusion. To solve this problem, canonical correlation analysis (CCA) (*e.g.*, *Haldorai & Ramu, 2020*; *Hotelling, 1936*) and kernel CCA are commonly used in multi-view clustering. Rasiwasia (e.g.,*Blaschko & Lampert ,2008*) proposed mean CCA and cluster CCA. Blaschko (*e.g.*, *Blaschko & Lampert, 2008*) projected the data across different views by KCCA and used k-means to cluster projected samples. We can find that the computational cost of the CCA and KCCA models is high (*e.g.*, *Wang, Li & Huang, 2017*). Besides, existing CCA-based clustering methods are only focus on mining the linear correlations. With the wide application of deep learning, the deep network has been applied in CCA model. Such as, Deep CCA (DCCA) (*e.g.*, *Andrew, Arora & Bilmes, 2013*), Deep Generalized CCA (DGCCA) (*e.g.*, *Benton, Khayrallah & Gujral, 2017*). However, DCCA model can only learn two views. Although DGCCA overcame the limitation of DCCA, it ignores mining the specific characteristics of multiple views. Subsequently, the newest multi-view representation learning model (MRL) (*e.g.*, *Zheng, Ge & Li, 2020*) based on DGCCA is proposed. This model is focus on mining the specific characteristics of inner-view and then learning the fusion representation based on the maximum correlation of multiple views. Recently, multi-view representation based on clustering has been viewed as the problem of learning a meaningful representation of data. Therefore, how to design model for multi-view clustering is an intriguing direction.

In summary, an effective mechanism is to learn a comprehensive meaningful representation with the discriminant characteristics in a single view and the correlation of multiple views. DCCA-based multi-view clustering methods are rarely used, but they have room for improvement with deep networks to mine nonlinear correlation and high-level fusion representations. With the aim of addressing the limitation of DCCA-based MVC methods, this paper proposes a unique novel framework called Dynamic Guided Metric

Representation Learning for Multi-View Clustering (DGMRL-MVC) that has not been investigated already in previous works on this topic.

The main contributions of this work can be summarized as follows:

● A multi-step enhanced representation is proposed for multi-view clustering, which consists of inter-intra learning, deep learning and latent space mapping. The proposed model can jointly learn a latent discriminated embedding.

● On the basis of learning intra-view class separability and inter-view consistency by Fisher Discriminant Analysis -with Hilbert–Schmidt Independence Criteria (FDA-HSIC) metric learning, a dynamic guided deep learning method is used that introduces location or direction information to enhance the single view representation.

● An ultimate common representation is obtained by generalized canonical correlation analysis (GCCA) model for multiple views.

● Experiments on four real-world multi-view datasets have validated the effectiveness of the proposed method for clustering tasks.

The rest of this paper is organized as follows: 'Related work' presents a review of related work. The next section introduces the proposed DGMRL-MVC model. 'Experiments' presents the datasets, experimental settings, and experimental results. Finally, 'Conclusions' draws conclusions.

# RELATED WORK

Related work on multi-view clustering methods can be divided into two categories: the common matrix framework (spectral clustering, subspace clustering, and non-negative matrix factorization clustering) and view fusion methods (multi-kernel clustering and DCCA-based methods). This section will review multi-view clustering methods from these two technological categories.

## Common matrix framework

Methods of this type have the commonality that they share the similar structure to combine multiple views.

### *Multi-view spectral clustering*

Multi-view spectral clustering assumes that all the views share the same or similar eigenvector matrices, in which co-training spectral clustering and co-regularized spectral clustering are the two representative methods.

(1) Co-training spectral clustering. These algorithms are investigated under the assumption of consensus among multiple views and trained alternately to maximize the consistency of the two distinct views. Three main assumptions are made: (1) each view is sufficient for the learning task; (2) the views are conditionally independent given the class labels; and (3) the objective functions export the same predictions for co-occurring features with high probability in both views. Overall, most co-training methods are semi-supervised learning.

(2) Co-regularized spectral clustering. Zhu (*e.g.*, *Zhu, Zhang & He, 2019*) proposed co-regularized approach. The main idea of this kind of method is to minimize the distinction

between the predictor functions of two views acting as one part of an objective function. They used graphic Laplacian eigenvectors to play a role much like that of predictor functions in a semi-supervised learning scenario. Inspired by the previous work (*e.g.*, *Zhu, Zhang & He, 2019*; *Chen, Huang & Wang, 2020*), automatically learned the weights of different views from data (e.g.,*Ye, Liu & Yin, 2016*) .

### *Multi-view subspace clustering*

In practice, multi-view data can be sampled from multiple subspaces. A subspace learning model can learn a new and unified representation or a latent space for multi-view data. The unified representation or latent space can then be directly used for the clustering task. In addition, a subspace clustering model can deal with high-dimensional data. The approach is to find the underlying subspaces and then to cluster. Wang (*e.g.*, *Wang, Lin & Wu, 2015*) proposed a model to measure correlation consensus in multiple views. Unlike Wang, Zhao (*e.g.*, *Zhao, Ding & Fu, 2017a*; *Zhao, Ding & Fu, 2017b*) used a deep semi-nonnegative matrix factorization to perform clustering. This kind of method focus on mining the inherent structure from multiple subspace of views and the clustering performance is heavily dependent on the affinity matrix. Therefore, some works used deep networks to learn the inter-view specific features based on classical subspace clustering methods. The newest multi-view representation learning model (MRL) (*e.g.*, *Zheng, Ge & Li, 2020*) is focus on mining the specific characteristics of inner-view and then learning the fusion representation based on the maximum correlation of multiple views. In all, how to mining the hidden difference of views is a research topic in recent years.

### *Multi-view non-negative matrix factorization clustering*

Given a non-negative matrix $X \in \mathbb{R}^{d \times n}$, non-negative matrix factorization (NMF) seeks two non-negative matrices $W \in \mathbb{R}^{d \times p}$ and $V \in \mathbb{R}^{p \times n}$, whose product the original matrix X:

$$X \approx WV^{T},$$

where W is the basis matrix and V the indicator matrix.

Due to its non-negativity constraints, NMF has emerged as a latent feature learning method. To combine multi-view information in an NMF framework, many variants of NMF have also been proposed. Kalayeh (*e.g.*, *Kalayeh, Idrees & Shah, 2014*) present a weighted extension of Multi-view NMF to address the aforementioned drawbacks. Liu (*e.g.*, *Liu, Wang & Lu, 2020*) used the cross entropy loss function to constrain the objective function better.

## View fusion methods

View fusion methods are also commonly used to learn multi-view information. However, their learning targets directly combine the views for clustering by different modes. Hence, they focus on how to combine multiple views.

### *Multi-view multi-kernel clustering*

Multi-kernel learning was originally used to combine views by means of a kernel and was widely used to deal with multi-view data. The common approach is to define a kernel for

each view and then combine these kernels in a convex combination (*e.g.*, *Sellami & Alaya, 2021*; *Lu, Liu & Wei, 2020*; *Nithya, Appathurai & Venkatadri, 2020*). Therefore, one main problem of this method is to choose kernel functions and optimal. A *k*-means analysis performed on kernel space that simultaneously found the best cluster labels, the cluster kernels, and the optimal combination of multiple kernels (*e.g.*, *Cui, Fern & Dy, 2007*). Liu (*e.g.*, *Liu, Ji & Glänzel, 2013*) extended *k*-means clustering into Hilbert space. Liu try to denote the data matrices as kernel matrices and then combined for fusion. In addition, consider the differences among views, methods with weighted combinations of kernels have also been studied.

However, multi-view data are incomplete. To address this issue, researchers integrated kernel imputation and clustering into a unified learning procedure (*e.g.*, *Monney, Zhan & Zhen, 2020*). Besides, other methods use a direct combination of features to perform multi-view clustering, like those in (*e.g.*, *Xu, Han & Nie, 2016*; *Nie, Li & Li, 2017*).

### Multi-view CCA-based methods clustering

For multi-view data clustering, it is reasonable to combine the data directly. However, it is hard to fuse different data types, and high dimensionality and noise are difficult to handle. The CCA method is used to combine the views directly and CCA-based techniques for multi-view clustering fusion. Chaudhuri (*e.g.*, *Chaudhuri, Kakade & Livescu, 2009*) projected the data into a lower-dimensional space using CCA and cluster. Then, he CCA-based methods are proposed, such as KCCA (*e.g.*, *Wang, Li & Huang, 2017*), DCCA(*e.g.*, *Andrew, Arora & Bilmes, 2013*), DGCCA (*e.g.*, *Benton, Khayrallah & Gujral, 2017*) and MRL(*e.g.*, *Zheng, Ge & Li, 2020*). Overall, although DCCA-based methods are used now in multi-view representation, this is a direction worthy of further multi-view clustering study using DCCA-based methods. In contrast to these methods, the representation will directly influence effectiveness on the clustering task. Thus, based on existing multi-view clustering learning methods, the intention is to design a novel model to learn a comprehensive meaningful data representation with the discriminant characteristics in a single view and correlation of all views.

In short, existing methods still has the following challenges: (1) Common matrix methods focus on learning a shared similar structure, they ignore the hidden difference of views. (2) View fusion methods considering the unique features of each view and the correlation of all views. However, existing methods strategy is two steps. The first step is self-representation of each view independently and the next step is focus on fusion views. The two-step collaboration is not well meet the clustering task.

## THE PROPOSED MODEL

### Motivation

Existing CCA-based multi-view clustering approaches that can deal with multi-view data learn the correlation of two views and perform a clustering task at the same time. Despite appealing performance, they still have some limitations. First, the quality of data representation is the key issue in MVC. And in real-world applications, directly fusing multi-view data is difficult and the effect is not good. Second, the discriminant characteristics in a

single view and the correlation of multiple views should all be considered. Considering this, a novel framework called Dynamic Guided Metric Representation Learning for Multi-View Clustering (DGMRL-MVC) is proposed in this paper, which can cluster multi-view data in a learned latent discriminated embedding space. Specifically, in the framework, the data representation can be enhanced by multi-steps. The model combines the effectiveness of FDA-HSIC and the dynamic routing learning of views embedded in the model. Hence, it can mine the class separability of single view, the consistence among different views, discriminant characteristics in a single view and a latent discriminated embedding for clustering task.

## Problem formulation

Given multi-view data observations, let $X_j \in \mathbb{R}^{d_j \times N}$ denote the $j$th view, where $d_j$ is the feature dimensionality of the $j$th view and $N$ is the number of samples.

**Definition 1.** (Mahalanobis distance). The Mahalanobis distance between two views $x_1$ and $x_2$ is defined as:

$$d_M^2(x_1, x_2) = \|x_1 - x_2\|_M^2 = (x_1 - x_2)^T M(x_1 - x_2), \tag{1}$$

where, the Mahalanobis matrix M is constrained to be symmetric positive-definite to ensure its validity.

## Framework
### Network architecture

Figure 1 illustrates the architecture of the proposed DGMRL-MVC model for multi-view clustering. This architecture effectively improves fusion representation and overcomes the limitations of traditional DCCA-based multi-view clustering learning. The model consists of three modules: inter-intra representation based on Fisher discriminant analysis and the Hilbert–Schmidt independence criteria, together called FDA-FISH metric learning; deep representation based on dynamic guided deep learning; shared representation based on deep generalized canonical correlation analysis. The first module learns intra-view separability and inter-view consistency. Then, as for dynamic guided deep learning, it is used to introduces location or direction information to enhance the single view representation. In the last sub-model, the fusion representation G of multiple views for the clustering task is created.

1) Inter-intra representation based on FDA-HSIC

The Fisher-HSIC Multi-View Metric Learning (FISH-MML) system is proposed in the work (*e.g.*, *Zhang, Liu & Liu, 2018*). This method is based on FDA and HSIC, which are simple, yet rather effective. Inspired by Zhang's work, an effort was made to add FDA-HSIC into the proposed framework for multi-view clustering representation learning.

**Intra-view separability.** The starting point is the definitions of between-class and total scatter matrices in the $v^{th}$ view:

$$S_b^{(v)} = \frac{1}{n} \sum_{j=1}^{m} n_j \left( \mu_j^{(v)} - \mu^{(v)} \right) \left( \mu_j^{(v)} - \mu^{(v)} \right)^T, \tag{2}$$

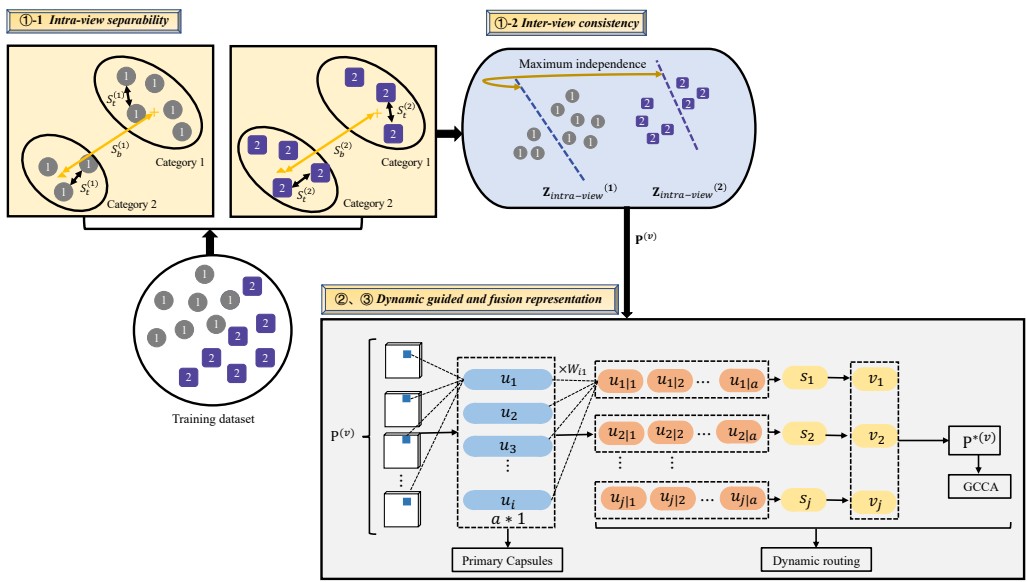

**Figure 1  Illustration of the proposed DGMRL-MVC model.**

$$S_t^{(v)} = \frac{1}{n}\sum_{i=1}^{n}(z_i^{(v)} - \mu^{(v)})(z_i^{(v)} - \mu^{(v)})^T,\tag{3}$$

$$\mu_j^{(v)} = \frac{1}{n_j}\sum_{i=1}^{n_j}z_i^{j(v)}, \mu^{(v)} = \frac{1}{n}\sum_{i=1}^{n_j}z_i^{(v)},\tag{4}$$

where $b$ is the abbreviation of between, $t$ is the abbreviation of total, $m$ is the number of classes, $n_j$ is the number of samples belonging to class $C_j$, $\mu_j^{(v)}$ are the sample means of the $v^{th}$ view for class $C_j$, and $\mu^{(v)}$ are the sample means of the $v^{th}$ view for all the views. $z_{ij^{(v)}}$ is the projected feature vector corresponding to $x_{ij^{(v)}}$ andis defined as follows:

$$z_{ij^{(v)}} = P^{(v)}x_i^{j(v)},\tag{5}$$

where the symmetric positive semi-definite matrix $M = \mathbf{P}^T\mathbf{P}$ and $\mathbf{P} \in \mathbb{R}^{k \times d}$ ($k \geq rank(M)$ is the new space.

The trace operator for $S_b^{(v)}$ and $S_t^{(v)}$ conditioned on $\mathbf{P}^{(v)}$ is then optimized by the following optimization function:

$$\max_{\{P^{(v)}\}_{v=1}^{V}}\sum_{v=1}^{V}Tr(\mathbf{S}_b^{(v)};\mathbf{P}^{(v)}) - \gamma\, Tr\left(\mathbf{S}_t^{(v)};\mathbf{P}^{(v)}\right),\tag{6}$$

where $\gamma$ is a tunable parameter to balance the two terms involved. Therefore, the above optimization function is focused on seeking metrics that jointly maximize separability.

**Inter-view consistency.** The next step is to explore the complementarity information from multi-views using HSIC (*e.g.*, *Gretton, Bousquet & Smola, 2005*). Based on HSIC, the

function can be defined as:

$$\text{HSCI}\left(\mathbf{Z}^{(v)}, \mathbf{Z}^{(w)}\right) = (n-1)^{-2}\text{tr}(\mathbf{K}_v \mathbf{H} \mathbf{K}_w \mathbf{H}), \tag{7}$$

$$\mathbf{K}^{(v)} = \mathbf{Z}^{(v)^T}\mathbf{Z}^{(v)} = \mathbf{X}^{(v)^T}\mathbf{P}^{(v)^T}\mathbf{P}^{(v)}\mathbf{X}^{(v)}, \tag{8}$$

$$\mathbf{K}^{(w)} = \mathbf{Z}^{(w)^T}\mathbf{Z}^{(w)} = \mathbf{X}^{(w)^T}\mathbf{P}^{(w)^T}\mathbf{P}^{(w)}\mathbf{X}^{(w)}, \tag{9}$$

where $\mathbf{K}^{(v)}$ and $\mathbf{K}^{(w)}$ are the Gram matrices from the two views parameterized by the projections $\mathbf{P}^{(v)}$ and $\mathbf{P}^{(w)}$. In the model, the dependency between $\mathbf{K}^{(v)}$ and $\mathbf{K}^{(w)}$ is enhanced by maximizing the HSIC function, and $\mathbf{H}$ is the Gram matrix that ensures zero mean in the feature space.

**Optimization.** The FDA-HSIC optimization function can be described as:

$$\max_{\{P^{(v)}\}_{v=1}^{V}} \sum_{v=1}^{V} Tr(\mathbf{S_b^{(v)}}; P^{(v)}) + \lambda_1 Tr(\mathbf{S_t^{(v)}}; P^{(v)}) + \lambda_2 \sum_{v \neq w} \text{HSIC}(\mathbf{P}^{(v)}\mathbf{X}^{(v)}, \mathbf{P}^{(w)}\mathbf{X}^{(w)})$$

$$= \max_{\{P^{(v)}\}_{v=1}^{V}} \sum_{v=1}^{V} Tr(P^{(v)}(\mathbf{A} + \lambda_1\mathbf{B} + \lambda_2\mathbf{C})P^{(v)^T})$$

$$= \max_{\{P^{(v)}\}_{v=1}^{V}} \sum_{v=1}^{V} tr(\mathbf{P}^{(v)}\mathbf{D}\mathbf{P}^{(v)^T})$$

$$s.t. \mathbf{P}^{(v)}\mathbf{P}^{(v)^T} = \mathbf{I}, \ v = 1, 2, \ldots, V, \tag{10}$$

where $\lambda_1 > 0 \ and \ \lambda_2 > 0$.

2) Deep representation based on dynamic guided learning

The output $\mathbf{P}^{(v)}$ of first submodel is fed into this submodel. $\mathbf{P}^{(v)}$ is the $v^{th}$ view. In this process, the learning of each view is independent. Inspired by Sabour's work (*e.g.*, *Sabour, Frosst & Hinton, 2017*), this submode adopt the dynamic guided mechanism to learn the location or direction information of view. Taking the first view $\mathbf{P}^{(1)}$ as example, we first divide $\mathbf{P}^{(1)}$ into several capsules $u_1, \ldots, u_{k1}$. Each capsule $u_{k1}$ by a weight matrix $W_{k_1 k_2}^*$ and get a prediction vector $\hat{o}_{k_2|k_1}$. $\hat{o}_{k_2|k_1}$ can be called deep learning mapping. Next, to represent the probability for the output vector, a nonlinear function squashing is applied to $s_{k_2}$. And the new capsule $s_{k_2}$ is weighted sum over $u_{k_1}$ with the coupling coefficient $c_{k_1 k_2}$.

$$\hat{o}_{k_2|k_1} = W_{k_1 k_2}^* u_{k_1}, \tag{11}$$

$$s_{k_2} = \sum_{k1} c_{k_1 k_2} \hat{o}_{k_2|k_1}, \tag{12}$$

$$c_{k_1 k_2} = \text{softmax}\left(p_{k_1 k_2}\right) = \frac{\exp(p_{k_1 k_2}^*)}{\sum_k \exp(p_{k_1 k}^*)}, \tag{13}$$

where the initial logits $p_{k_1 k_2}^*$ is the log prior probabilities that capsule $k_2$ to $k_1$.

Finally, the output of the first view is $\mathbf{P}^{*(1)}$, it is denoted as: $\mathbf{P}^{*(1)} = [\mathbf{v}_1, \mathbf{v}_2, \ldots\ldots, \mathbf{v}_{k_2}]$, where $\mathbf{v}_{k_2} = [v_1, \ldots, v_j]$. The discriminant characteristics of single view is learned:

$$v_{k_2} = \text{squash}(s_{k_2}) = \frac{\|s_{k_2}\|^2}{1 + \|s_{k_2}\|^2} \frac{s_{k_2}}{\|s_{k_2}\|}, \tag{14}$$

3) Shared representation based on deep generalized canonical correlation analysis

This sub-model learns the ultimate common fusion representation $\mathbf{G}$ from all views. In this process, minimized loss function as follows:

$$\underset{U_v \in \mathbb{R}^{d_v \times r}, G \in \mathbb{R}^{r \times N}}{\text{minimize}} \sum_{v=1}^{V} \left\| \mathbf{G} - \mathbf{U}_v^{\mathbf{T}} \mathbf{P}^{*(v)} \right\|_{\mathbf{F}}^2,$$

$$\text{subject to } \mathbf{G}\mathbf{G}^{\mathbf{T}} = \mathbf{I_r}, \tag{15}$$

where $\mathbf{P}^{*(v)} \in \mathbb{R}^{d_v \times N_v}$ is latent discriminated embedding of the $v^{th}$ view. Then, all $P_v^*$ spliced into a matrix $\mathbf{G}$.

### Training

In this model, the scaled empirical co-variance matrix $\mathbf{C}_{vv}$ of the $v^{th}$ network is calculated by $\mathbf{C}_{vv} = \mathbf{P}^{*(v)}(\mathbf{P}^{*(v)})^{\mathbf{T}} \in \mathbb{R}^{d_v * d_v}$. Where $\mathbf{P}^{*(v)} = \mathbf{P}^{*(v)\mathbf{T}} \mathbf{C}_{vv}^{-1} \mathbf{P}^{*(v)} \in \mathbb{R}^{N*N}$ is the corresponding projection matrix. In optimization process, we consider maximize the sum of correlations between fusion representation $\mathbf{G}$ and single view. The reconstruction error as follows:

$$\sum_{v=1}^{V} \left\| \mathbf{G} - \mathbf{U}_v^{\mathbf{T}} \mathbf{P}^{*(v)} \right\|_{\mathbf{F}}^2 = \sum_{v=1}^{V} \left\| \mathbf{G} - \mathbf{G}\mathbf{P}^{*(v)\mathbf{T}} \mathbf{C}_{vv}^{-1} \mathbf{P}^{*(v)} \right\|_{\mathbf{F}}^2$$

$$= r\mathbf{J} - \text{Tr}(\mathbf{G}\mathbf{M}\mathbf{G}^T), \tag{16}$$

where the positive semi-definite matrix of each view is $\mathbf{M} = \sum_{v=1}^{V} \mathbf{P}_v$, $r$ is the top rows of $\mathbf{M}$. To learn the correlation of every pair of views, we stack them in a $j * j$ matrix, it is defined as $\mathbf{J}$.

The derivative of loss function is:

$$\frac{\partial \mathbf{L}}{\partial \mathbf{P}_v^*} = 2\mathbf{U}_v G - 2\mathbf{U}_v \mathbf{U}_v^{\mathbf{T}} \mathbf{O}_v^*, \tag{17}$$

where $L = \sum_{i=1}^{r} \lambda_i(\mathbf{M})$.

## EXPERIMENTS

### Datasets

We select four datasets come from real-world to verify model. The views of these datasets include network, text, IDs and image.

- **Football**. This dataset is collected as previously described in *Greene & Cunningham (2013)*. It is the Twitter active data of 248 English Premier Leaguefootball players and clubs. This dataset contains nine views and 20 class labels.

- **Wikipedia**. This datasetis collected as previously described in *Rasiwasia, Mahajan & Mahadevan (2014)* & *Pereira et al. (2014)*. It come fromWikipedia's featured articles. This dataset contains two views and 10 categories.
- **Handwritten**. This dataset is collected through https://archive.ics.uci.edu/ml/datasets/Multiple+Features. It contains six views and 10 categories.
- **Pascal**. This dataset is collected as previously described in C.Rashtchian et al. (2010). It contains six views and 20 categories.

## Experimental settings

In the experiments, we choose 80% of data randomly to train and other 20% to test. In FDA-HSCI, the parameter tuning was as in Zhang's work (*e.g.*, *Zhang, Liu & Liu, 2018*). The maximum training iterations and learning rate $\eta$ as in *Benton, Khayrallah & Gujral (2017)* and *Chandar, Khapra & Larochelle (2015)*.

## Evaluation metrics

The effect of our model is evaluated by comparing it with baselines in terms of clustering precision, recall, F1, RI, Normalized Mutual Information(NMI) and Silhouette_score. These evaluation measures are defined as follows:

$$precision = \frac{TP}{TP+FP}, \tag{18}$$

$$recall = \frac{TP}{TP+FN}, \tag{19}$$

$$F1 = \frac{2PR}{P+R}, \tag{20}$$

$$RI = \frac{TP+TN}{TP+FP+FN+TN}, \tag{21}$$

where TP and FP are positive and negative samples predicted as positive classes, TN and FN are positive and negative samples predicted as negative classes.

$$NMI(X;Y) = \frac{2I(X;Y)}{H(X)+H(Y)}, \tag{22}$$

where $I(X;Y) = \sum_x \sum_y p(x,y) \log \frac{p(x,y)}{p(x)p(y)}, H(X) = -\sum_i p(x_i) \log p(x_i)$.

The Silhouette_score for a sample is:

$$Silhouette\_score = \frac{b-a}{\max(a,b)}, \tag{23}$$

where $b$ is the distance between a sample and the nearest cluster of which the sample is not a member. Values close to 1 indicate that a sample has been assigned to a better cluster because a different cluster is more similar.

## Baselines

The following methods were compared in the experiments described in this section:

- **DCCA**: The method realizes the deep learning applied in CCA. However, DCCA can only learn nonlinear mapping of two views (*e.g.*, *Andrew, Arora & Bilmes, 2013*).

- **DGCCA:** Although this model overcame the view number limitation of DCCA, it ignores mining the specific characteristics of multiple views (*e.g.*, *Benton, Khayrallah & Gujral, 2017*).

- **FISH-MML**: A Fish-HSIC multi-view metric learning method, which is simple, yet rather effective. The model can learn intra-view separability and inter-view correlation (*e.g.*, *Zheng, Ge & Li, 2020*).

- **Dynamic guided representation-MVC:** To better verify the validity of the model with FDA-FISH, only the dynamic guided representation was added to the proposed model (*e.g.*, *Zheng, Ge & Li, 2020*).

Table 1 shows the differences among these methods. In summary, the effectiveness of the proposed DGMRL-MVC model was directly evaluated by comparing it with newest DCCA-based multi-view learning methods. The significance of the dynamic guided representation mechanism in the proposed model was further verified by comparison with FISH-MML. To provide a further comprehensive analysis of the stability and performance of the proposed multi-view representation learning model, the dynamic guided representation-MVC was also chosen as a baseline.

## Results

The experimental results answered the following questions:

### (RQ1) How good is the performance improvement of the proposed model with dynamic guided representation? (DGMRL-MVC vs. FISH-MML)

FISH-MML provided the best multi-view metric learning with FDA-FISH. The proposed DGMRL-MVC model was first compared with FISH-MML. The results showed that DGMRL-MVC significantly outperformed FISH-MML on all four datasets for the multi-view clustering task. Table 2 presents the comprehensive comparison results. It is obvious that the proposed model achieved the best performance on nearly all datasets under all metrics. Take the Handwritten dataset with six views as an example, the improvement of DGMRL-MVC over FISH-MML was about 42.2%, 60.4%, 46%, 60.3% and 19.5% in terms of precision, recall, F1, RI, and NMI respectively. This proved that the dynamic guided representation used in the proposed method is effective for multi-view clustering. Our model has been enhanced in two aspects, *i.e.,* inter-intra relations and dynamic routing learning embedded into each view self-representation, and fusion representation with maximum correlation.

### (RQ2) How good is the performance of DGMRL-MVC compared with other DCCA-based methods?

The latent DCCA-based representation was compared with the proposed algorithm on a clustering task. DCCA and DGCCA were chosen as baselines. In the dynamic guided representation-MVC model, FDS-FISH was not present, and only dynamic guided

**Table 1** The differences among baselines.

| Method | Type | Dynamic guided representation | FDA-HSIC | Optimization |
|---|---|---|---|---|
| DCCA | Deep, 2-view | × | × | $(\theta_1^*, \theta_2^*) = \arg\max_{(\theta_1,\theta_2)}(f_1(X_1;\theta_1), f_2(X_2;\theta_2))$ |
| DGCCA | Deep, multi-view | × | × | $\underset{U_j \in \mathbb{R}^{d_j \times r}, G \in \mathbb{R}^{r \times N}}{\text{Minimize}} \sum_{j=1}^{J} \left\| G - U_j^T f_j(X_j) \right\|_F^2$ |
| FISH-MML | Deep, multi-view | × | ✓ | $max_{\{M^{(v)}\}_{v=1}^{V}} S\left(\{M^{(v)}\}_{v=1}^{V}\right) + \lambda C(\{M^{(v)}\}_{v=1}^{V})$ |
| Dynamic guided representation-MVC | Deep, multi-view | ✓ | × | $\underset{U_j \in \mathbb{R}^{d_j \times r}, G \in \mathbb{R}^{r \times N}}{\text{Minimize}} \sum_{j=1}^{J} \left\| G - U_j^T O_j^* \right\|_F^2$ |
| DGMRL-MVC | Deep, multi-view | ✓ | ✓ | $\underset{U_v \in \mathbb{R}^{d_v \times r}, G \in \mathbb{R}^{r \times N}}{\text{minimize}} \sum_{v=1}^{V} \left\| G - U_v^T P^*(v) \right\|_F^2$ |

**Table 2** Improvement performance of dynamic guided representation on different views. The best results are highlighted in bold.

| Dataset | Method | Number of views | Precision | Recall | F1 | RI | NMI |
|---|---|---|---|---|---|---|---|
| Handwritten | FISH-MML | 6 (full views) | 0.09961 | 0.11099 | 0.10493 | 0.11100 | 0.09643 |
| | DGMRL-MVC | | **0.14166** | **0.17800** | **0.15315** | **0.17800** | **0.11521** |
| | FISH-MML | 2 views | 0.17843 | 0.20600 | 0.18586 | 0.20600 | 0.21906 |
| | DGMRL-MVC | | **0.25699** | **0.22100** | **0.22521** | **0.22100** | **0.32698** |
| | FISH-MML | 3 views | 0.09092 | 0.10100 | 0.09569 | 0.10100 | 0.25037 |
| | DGMRL-MVC | | **0.17813** | **0.14300** | **0.13452** | **0.14300** | **0.37442** |
| Wikipedia | FISH-MML | 2 (full views) | 0.08957 | 0.10149 | 0.09313 | 0.10032 | 0.51780 |
| | DGMRL-MVC | | **0.10491** | **0.12321** | **0.10398** | **0.13438** | **0.53230** |
| Football | FISH-MML | 9 (full views) | 0.05679 | 0.05885 | 0.01761 | 0.06048 | 0.22956 |
| | DGMRL-MVC | | **0.05900** | **0.06248** | **0.05575** | **0.06452** | **0.28367** |
| | FISH-MML | 6 views | **0.11914** | 0.04903 | 0.03745 | 0.05645 | **0.28422** |
| | DGMRL-MVC | | 0.05619 | **0.06506** | **0.05848** | **0.06452** | 0.27343 |
| Pascal | FISH-MML | 6 (full views) | 0.07616 | 0.06700 | 0.06606 | 0.06700 | 0.23351 |
| | DGMRL-MVC | | **0.08315** | **0.07500** | **0.07342** | **0.07500** | **0.48847** |
| | FISH-MML | 2 views | 0.06266 | 0.05600 | 0.05094 | 0.05600 | 0.18011 |
| | DGMRL-MVC | | **0.08313** | **0.06800** | **0.06448** | **0.06800** | **0.45506** |

representation and generalized canonical correlation analysis were added. As shown in Table 3, the results showed that DGMRL-MVC significantly outperformed DGCCA and DCCA on all datasets. In addition, the performance using the proposed discriminated representation was generally better than that using dynamic guided representation-MVC. In all, our model achieved comprehensive results because our model adds FDA-FISH and dynamic guided module that distills more separable intra-view and specific intrinsic features, resulting in a high-quality latent discriminated embedding.

**Table 3  Comparison with latest methods with multiple views.** The best results are highlighted in bold.

| Dataset | Method | Number of views | Precision | Recall | F1 | RI | NMI |
|---|---|---|---|---|---|---|---|
| Handwritten | DCCA | | 0.10111 | 0.10000 | 0.10047 | 0.10000 | **0.57354** |
| | DGCCA | | 0.08222 | 0.05400 | 0.06244 | 0.10800 | 0.54150 |
| | FISH-MML | 6 | 0.09961 | 0.11099 | 0.10493 | 0.11100 | 0.09643 |
| | Dynamic guided representation-MVC | (full views) | 0.09835 | 0.11650 | 0.10402 | 0.11650 | 0.52245 |
| | DGMRL-MVC | | **0.14166** | **0.17800** | **0.15315** | **0.17800** | 0.11521 |
| Wikipedia | DCCA | | 0.09245 | 0.08467 | 0.08289 | 0.09090 | 0.50819 |
| | DGCCA | | 0.10329 | 0.10567 | 0.10267 | 0.10400 | 0.54222 |
| | FISH-MML | 2 | 0.08957 | 0.10149 | 0.09313 | 0.10032 | 0.51780 |
| | Dynamic guided representation-MVC | (full views) | 0.16876 | 0.14224 | 0.13550 | 0.17027 | 0.59470 |
| | DGMRL-MVC | | **0.19480** | **0.18393** | **0.22795** | **0.18110** | **0.59851** |
| Football | DCCA | | 0.05625 | 0.04909 | 0.05200 | 0.05241 | 0.21037 |
| | DGCCA | | 0.01404 | 0.01885 | 0.01603 | 0.02016 | 0.21256 |
| | FISH-MML | 9 | 0.05679 | 0.05885 | 0.01761 | 0.06048 | 0.22956 |
| | Dynamic guided representation-MVC | (full views) | **0.10516** | 0.06449 | 0.05430 | 0.06451 | 0.26470 |
| | DGMRL-MVC | | 0.07540 | **0.06867** | **0.06647** | **0.07258** | **0.28367** |
| Pascal | DCCA | | 0.02123 | 0.04200 | 0.02814 | 0.04200 | 0.03363 |
| | DGCCA | | 0.04716 | 0.06600 | 0.05351 | 0.06600 | 0.09261 |
| | FISH-MML | 6 | 0.07616 | 0.06700 | 0.06606 | 0.06700 | 0.23351 |
| | Dynamic guided representation-MVC | (full views) | 0.06619 | 0.07200 | 0.04782 | 0.07200 | 0.15842 |
| | DGMRL-MVC | | **0.08315** | **0.07500** | **0.07342** | **0.07500** | **0.48847** |

**(RQ3) Is the discriminated fusion representation good on the clustering task?**
As shown in Tables 4 and 5, the visualization is consistent with the clustering results. In this experiment, the *Silhouette_score* was also chosen as the evaluation indicator. Tables 4 and 5 reveal the advantage of the proposed model when the dynamic guided and FDA-HSIC models were used simultaneously. The *Silhouette_score* for clustering was higher than that for the FISH-MML model. Besides, the *Silhouette_score* of the full view method was higher than that of the 2-view method, which proved the necessity of the multi-view fusion representation. Our model has better learning performance by using full views, and can effectively mine the features of multiple views. The result empirically proves that clustering with full-view is more robust that with 2-view.

**(RQ4) How does performance vary w.r.t. the parameter of n_clusters?**
Parameter analysis was conducted on the *Silhouette_score* parameter by varying the number of clusters in [2,3,5,7,8,9,10] (Handwritten and Wikipedia datasets) and [2,5,8,10,13,15,18,20] (Football and Pascal datasets). Figures 2, 3, 4 and 5 plots the results in terms of *Silhouette_score* when different numbers of clusters were used on the four datasets. The proposed DGMRL-MVC showed the best performance on the other datasets. However,

**Table 4   Visualization of clustering result with FISH-MML on the Pascal dataset.** The best results are highlighted in bold.

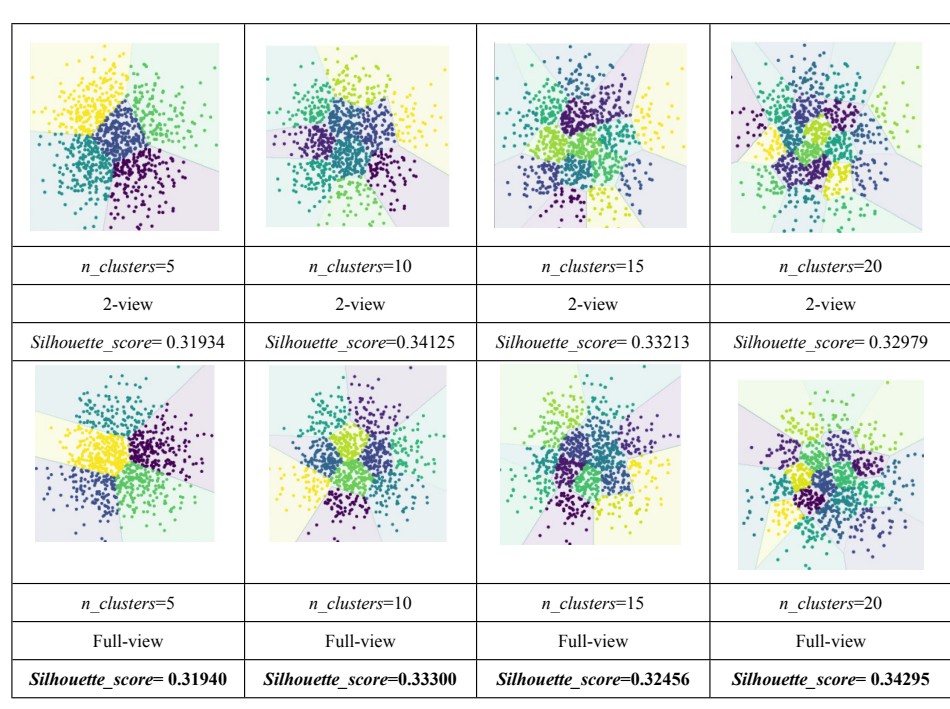

DGMRL-MVC slightly worse than the other methods on the Handwritten dataset. Further, combined with the previous experimental results, we analysis the reason of Fig. 2 result. First, this dataset is different from other datasets, its views are six types of features on same picture, *i.e.,* Fourier coefficients of the character shapes, profile correlations, Karhunen-love coefficients, pixel averages in $2\times 3$ windows, Zernike moment, morphological. Due to the large difference and weak correlation between the original data views, our model weakens the original relationship of views. The value of the *Silhouette_score* reflects the distance of samples in same cluster. In detail, although clustering evaluation metrics of our model is higher than the baselines, the sample is closer to the boundary of the cluster in this kind of dataset.

### (RQ5) Ablation experiment

Finally, ablation experiments were used to further verify the effectiveness of the model. We compared the differences of model learning performance under three steps respectively. Specifically, Step-A is inter-intra representation, Step-B is fusion representation directly, Step-C is dynamic routing and fusion representation. From the results of Fig. 6, we can find that performance of Step-B is lower than Step-C. The main reason is that fusion learning is only aimed at the maximum correlation between views, lacking difference learning of views, which is not conducive to the feature expression for clustering tasks. At the same time, the performance difference between Step-B and Step-A is not big, and only some indicators are slightly higher than Step-A, indicating that in the process of fusion learning,
**Table 5 Visualization of clustering result with DGMRL-MVC (ours) on the Pascal dataset.** The best results are highlighted in bold.

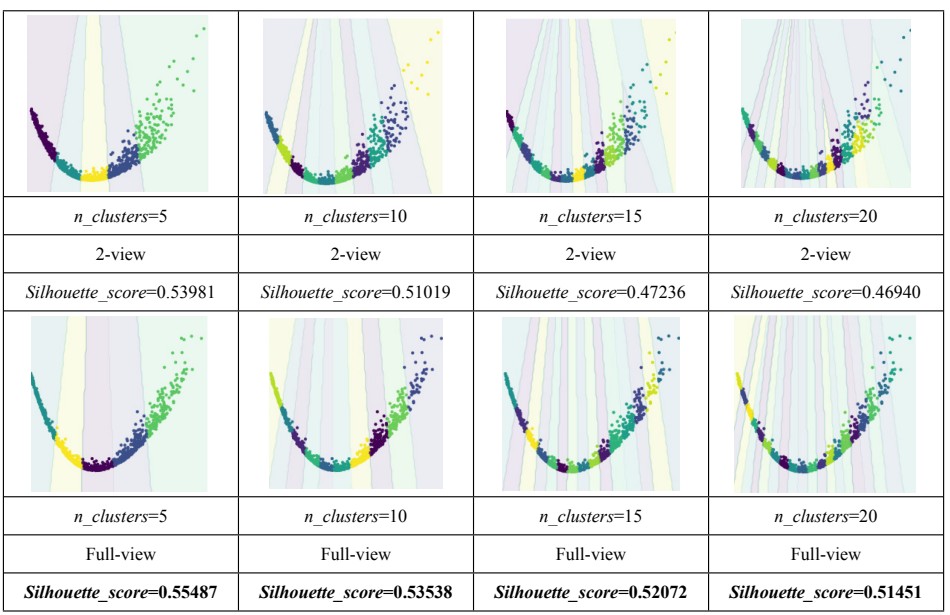

| | | | |
|---|---|---|---|
| *n_clusters*=5 | *n_clusters*=10 | *n_clusters*=15 | *n_clusters*=20 |
| 2-view | 2-view | 2-view | 2-view |
| *Silhouette_score*=0.53981 | *Silhouette_score*=0.51019 | *Silhouette_score*=0.47236 | *Silhouette_score*=0.46940 |

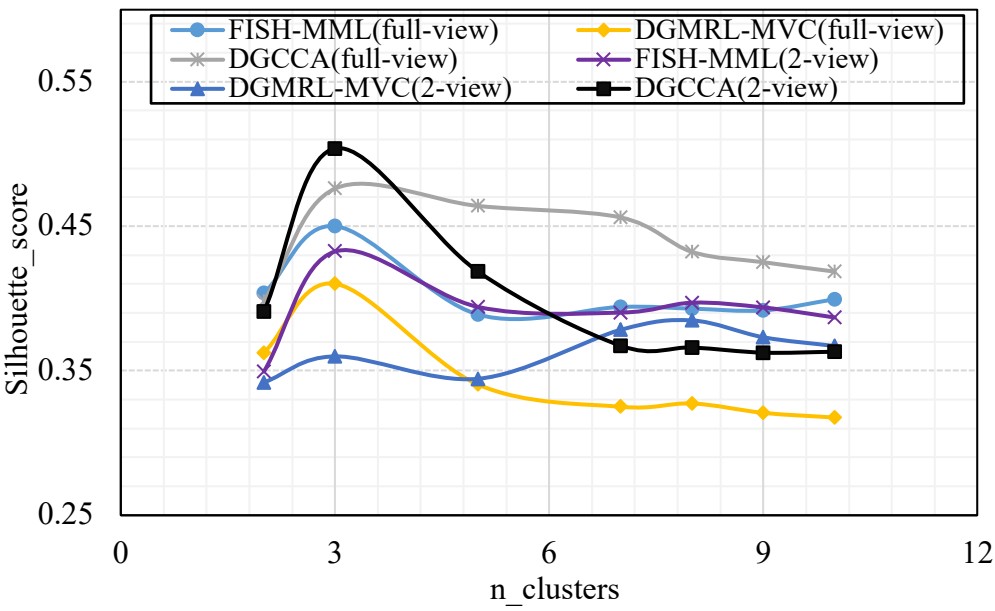

| | | | |
|---|---|---|---|
| *n_clusters*=5 | *n_clusters*=10 | *n_clusters*=15 | *n_clusters*=20 |
| Full-view | Full-view | Full-view | Full-view |
| ***Silhouette_score*=0.55487** | ***Silhouette_score*=0.53538** | ***Silhouette_score*=0.52072** | ***Silhouette_score*=0.51451** |

**Figure 2 Parameter analysis on *n_clusters* in terms of *Silhouette_score* on the Handwritten dataset.**

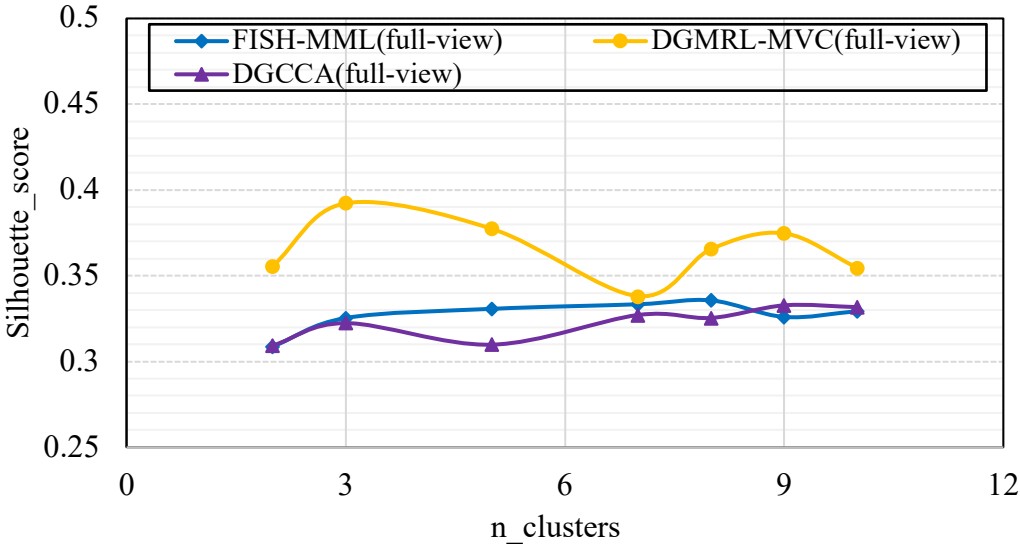

**Figure 3** Parameter analysis on *n_clusters* in terms of *Silhouette_score* on the Wikipedia dataset.

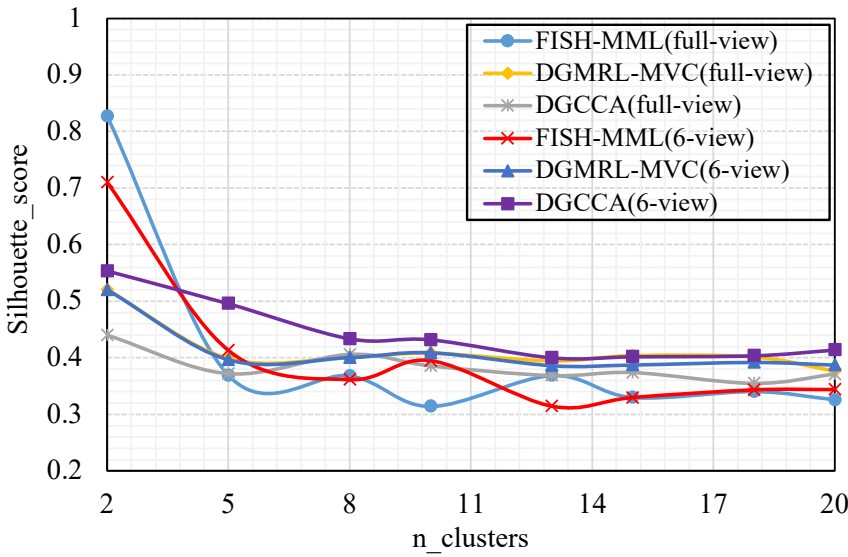

**Figure 4** Parameter analysis on *n_clusters* in terms of *Silhouette_score* on the Football dataset.

the specific intrinsic features learning with dynamice routing plays the most obvious role. Our method result is higher than other sub-models, which proves that on the basic of strengthening the learning view relationship, adding subspace fusion characterization can effectively improve the clustering performance of model.

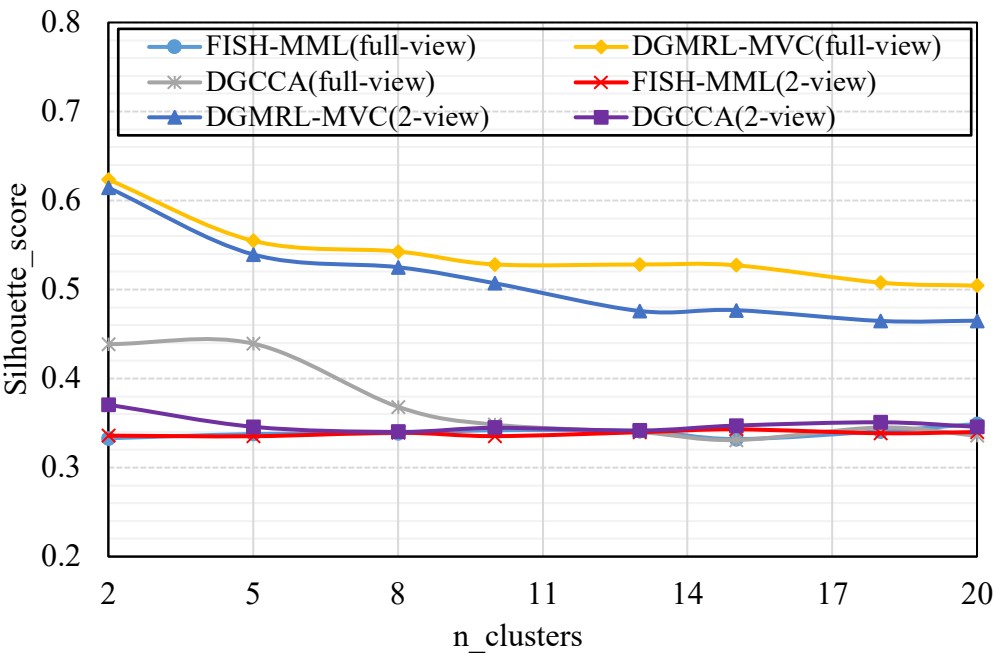

**Figure 5** Parameter analysis on *n_clusters* in terms of *Silhouette_score* on the Pascal dataset.

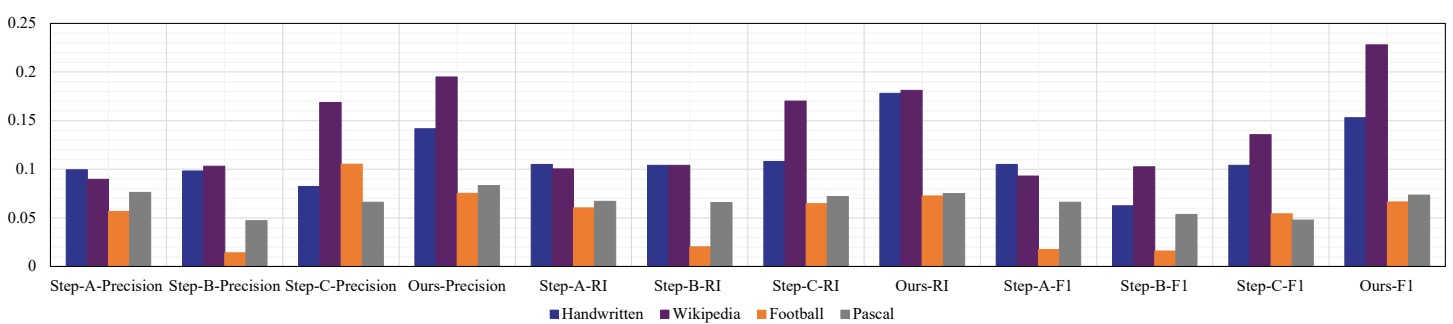

**Figure 6** Ablation experiments on four multi-view datasets.

## CONCLUSIONS

This paper has proposed a novel framework called Dynamic Guided Metric Representation Learning for Multi-View Clustering (DGMRL-MVC). The proposed method could cluster multi-view data in a learned latent discriminated embedding space with multiple views by multi-step enhanced representation. Within this framework, intra-view class separability and the complex correlations among different views were explored using FDA-HSIC. Then the direction information is added to enhance the single view representation by dynamic routing deep learning mechanism. Finally, the common discriminated representation was obtained by generalized canonical correlation analysis for multiple views and directly

improved MVC performance. Experiments on four multi-view datasets have demonstrated the effectiveness of the proposed method for clustering tasks.

Multi-view data exits everywhere and there are still many challenging problems, such as data alignment from heterogeneous sources (*e.g.*, *Wang, Zheng & Li, 2017*), unified multi view fusion model design and optimization (*e.g.*, *Tang et al., 2020*), and et al. In the future, we will do more work on multi view data fusion.

### Funding

This work was supported by the National Natural Science Foundation of China (No.61872260), the Shanxi Basic Research Program (free exploration) Project (No.20210302124551), the Scientific and Technological Innovation Programs of Higher Education Institutions in Shanxi (No. 2020L0735), the Institute-Level Research Fund Project of Shanxi Energy College (No.SY-2018004), the Shanxi Institute of Energy excellent sharing course project (No.NJP202007), the Shanxi Institute of Energy teaching reform and innovation project (No.NJ202016), and the Shanxi Institute of Energy teaching reform and innovation project (No.NJP202009). The funders had no role in study design, data collection and analysis, decision to publish, or preparation of the manuscript.

### Grant Disclosures

The following grant information was disclosed by the authors:
The National Natural Science Foundation of China: No.61872260.
Shanxi Basic Research Program (free exploration) Project: No.20210302124551.
the Scientific and Technological Innovation Programs of Higher Education Institutions in Shanxi: No. 2020L0735.
the Institute-Level Research Fund Project of Shanxi Energy College: No.SY-2018004.
Shanxi Institute of Energy excellent sharing course project: No.NJP202007.
Shanxi Institute of Energy teaching reform and innovation project: No.NJ202016.
Shanxi Institute of Energy teaching reform and innovation project: No.NJP202009.

### Competing Interests

The authors declare there are no competing interests.

### Author Contributions

- Tingyi Zheng Yilin Zhang conceived and designed the experiments, performed the experiments, analyzed the data, performed the computation work, prepared figures and/or tables, authored or reviewed drafts of the paper, and approved the final draft.
- Tingyi Zheng, Yuhang Wang performed the experiments, authored or reviewed drafts of the paper, and approved the final draft.

### Data Availability

The datasets are available in the Supplementary Files.

## Supplemental Information

Supplemental information for this article can be found online at http://dx.doi.org/10.7717/peerj-cs.922#supplemental-information.

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
