# Peer review of "Dynamic guided metric representation learning for multi-view clustering"

_PeerJ Computer Science, doi:10.7717/peerj-cs.922_

## Round 0.1 · original submission · Minor Revisions

The paper was evaluated by three reviewers who provided thorough comments and feedback. All reviewers find that the paper contains effort to conduct point estimation tasks. The reviewers give some additional comments and suggestions on how the paper could be improved. Please see their full comments for that. I suggest the authors highlight all modifications in the revised version and answer point-by-point the reviewer's comments. It is also expected that the authors can clearly highlight a unique novel aspect that has not been investigated already in previous works on this topic.

·

Basic reporting

This paper proposes a framework called Dynamic Guided Metric Representation Learning for Multi-View Clustering, which is clearly and unambiguously expressed. The literature references is sufficient. The structure, figures and tables are profession. Experimental datasets are shared. However, the writing could be further improved. More explanations about the expeirmental comparison should be added, besides reporting the best resuts. Moreover, well-designed ablation studies could be further enhance the effectiveness of the proposed framework.
Therefore, I recommend this paper to be accepted after a minor revision.

Experimental design

Overall, the experiments are well-designed and meaningful. Several comments are listed as follows:
1. More explanations about the expeirmental comparison should be added, besides reporting the best resuts.
2. Well-designed ablation studies could be further enhance the effectiveness of the proposed framework.

Validity of the findings

no

Additional comments

The written of the manuscript should be carefully checked and improved. Several examples of typos are listed here:
1. In the abstract, make sure whether "... 1rt " is a mistake;
2. Line 259-260, and et al.

Reviewer 2 ·

Basic reporting

This paper proposed a method with multiple learning steps to cluster multi-view data including inter-intra learning, dynamic guided deep learning and shared representation. It proposed an effective mechanism that considers the discriminant characteristics in a single view and the correlation between views simultaneously. Specially, it combines the effectiveness of FDA-HSIC and dynamic routing learning, and solves issues that occurs with existing methods. In the end, experiments on four datasets demonstrated the effectiveness of the proposed method for clustering tasks.

This paper needs to be improved in the following aspects:
1. In the figure 2 of experiments part, the silhouette_score of this method is lower than the baselines, authors should give more detailed analysis on this.

2. There are some grammatical and spelling mistakes that need to polish.
For example: what are the variables t and b in formula 2 and 3? Is “Deep representation based on and dynamic guided deep learning ” correct ?

3. In Section 4.3, the reference of baseline algorithms should be added.

4. In related work, a paragraph to summarize the main technical challenges of related work should be added.

Experimental design

Experiments are well desined. Datasets and experimental results are presented clear.

Validity of the findings

Conclusion are well stated.

Reviewer 3 ·

Basic reporting

This paper presents an interesting study on multi-view clustering method. In particular, the authors have proposed a novel framework called DGMRL-MVC, which can cluster multi-view data in a learned latent discriminated embedding space. The data representation can be enhanced by multi-steps. And the approach is proposed to address issues that arise with existing methods, through considering the discriminant characteristics in a single view and the correlation of multiple views rather than using this information from a single view. The effectiveness of the proposed approach has been validated experimentally by compared baseline methods on four multi-view datasets. The results show some improvements of the performance in terms of precision, recall, F1, RI, NMI and Silhouette_score, in comparison with existing methods.

This paper needs to be improved in the following aspects:
1)The proposed approach is learning the embedding space for clustering. Therefore, authors should add some relevant details on multi-view subspace clustering in the Section 2.1.2.
2)Please check the last sentence in the Section 2.2.2 and Section 3.3.1.

Experimental design

1)In Experiments, the authors state that the evaluation measures include precision, recall, F1, RI, NMI. In order to make the effect clearer to reader,authors should add the formulas.
2)In the Section 4.3, Table 1 shows the differences among baselines. Authors should add the more relevant details (e.g., optimization function).

Validity of the findings

No

Additional comments

1)Please check the abbreviation of the model in Fig.2 and Fig.3.
2)Please standardize the punctuation after the formula.

---

## Round 0.2 · accepted · Accept

The manuscript has been reevaluated and well revised to address the reviewers' comments. The statistical analysis is performed to the technical standard required for publication. Therefore, I recommend an acceptance.

Reviewer 2 ·

Basic reporting

In this revision, authors have answered the question I asked in last review. I have no further question for this revision.

Experimental design

authors have well explained the experimental results in Figure 2.

Validity of the findings

Same as last review.

Reviewer 3 ·

Basic reporting

OK.

Experimental design

Ok.

Validity of the findings

Ok,

Additional comments

Ok